# Synergistic Effect of Polymorphs in Doped NaNi$_{0.5}$Mn$_{0.5}$O$_2$ Cathode Material for Improving Electrochemical Performances in Na-Batteries

Francesco Leccardi, Davide Nodari , Daniele Spada, Marco Ambrosetti and Marcella Bini *

Chemistry Department, University of Pavia, Viale Taramelli 16, 27100 Pavia, Italy;
francesco.leccardi01@universitadipavia.it (F.L.); davide.nodari02@universitadipavia.it (D.N.);
daniele.spada01@universitadipavia.it (D.S.); marco.ambrosetti01@universitadipavia.it (M.A.)
* Correspondence: marcella.bini@unipv.it

**Abstract:** Layered NaNi$_{0.5}$Mn$_{0.5}$O$_2$, employed as cathode materials in sodium ion batteries, is attracting interest due to its high working potential and high-capacity values, thanks to the big sodium amount hosted in the lattice. Many issues are, however, related to their use, particularly, the complex phase transitions occurring during sodium intercalation/deintercalation, detrimental for the structure stability, and the possible Mn dissolution into the electrolyte. In this paper, the doping with Ti, V, and Cu ions (10% atoms with respect to Ni/Mn amount) was used to stabilize different polymorphs or mixtures of them with the aim to improve the capacity values and cells cyclability. The phases were identified and quantified by means of X-ray powder diffraction with Rietveld structural refinements. Complex voltammograms with broad peaks, due to multiple structural transitions, were disclosed for most of the samples. Ti-doped sample has, in general, the best performances with the highest capacity values (120 mAh/g at C/10), however, at higher currents (1C), Cu-substituted sample also has stable and comparable capacity values.

**Keywords:** NaNi$_{0.5}$Mn$_{0.5}$O$_2$; sodium batteries; cyclic voltammetry; charge–discharge measurements; ex-situ XRPD; doping

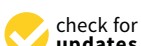



## 1. Introduction

Layered oxides are, nowadays, one of the most promising category of cathode materials for sodium ion batteries [1–3]. They offer a large variety of possible compositions: for the compounds with Na$_x$MeO$_2$ formula, both Na content (up to 1) and Me ion (transition element) can be varied, also allowing mixture of transition elements in several ratios [1]. Furthermore, different piling up of the MeO$_6$ layers along the crystallographic $c$ axis can be formed, originating O and P type structures, as named by Delmas, depending on the sodium coordination (octahedral or prismatic) [4]. NaMeO$_2$ (Me = Fe, Ni, Mn, Cr, Co and others) could be crystallized in the most common O3 type structure with AB-CA-BC layers packing and Na ions in octahedral sites. Another energetically favored polymorph, as demonstrated by theoretical calculations, is P2 type, with AB-BA layers and prismatic sodium [5]. The partial removal of sodium causes the stabilization of other structures, particularly, P3 or O2, as well as monoclinic distortions (marked by the symbol ′ after the capital letter P or O), which can possibly form because of intercalation/deintercalation of sodium during cell functioning [6,7]. Na[Ni$_{0.5}$Mn$_{0.5}$]O$_2$ with O3 type structure has the inherent advantages of high working potential of Ni$^{2+/3+/4+}$ redox couples and high capacity due to Mn ions [8]. More importantly, compared with sodium-deficient P2-type materials, this compound has higher sodium content to provide more cyclable sodium and high initial Coulombic efficiency, which play a pivotal role in practical sodium-ion full cell systems. It has a theoretical capacity of about 200 mAh g$^{-1}$ based on Ni$^{2+/4+}$ redox couple with a maximum potential of ~4.0 V, high electrochemical activity and reduced cost [9].

The theoretical capacity can increase up to 240 mAh g$^{-1}$ when the potential window is extended to 4.5 V.

Differently from its lithium analogue, it does not suffer from sodium diffusion, because antisite defects are hampered due to the large difference between Na$^+$ and Ni$^{2+}$ ionic radii. Nevertheless, some issues need to be solved. First, the Jahn–Teller effect of Mn$^{3+}$ ions and the multiple phase transformations during the sodium deintercalation/intercalation processes (e.g., from O3 to O′3, P3, and P′3 phases) have a detrimental influence on the cycling performance [8,9]. In addition, the poor air stability raises difficulties during the synthesis processes. In order to remove these barriers, doping has been demonstrated to be an efficient solution [10–18].

It has been ascertained that by changing the stoichiometry, different polymorphs can be stabilized, as well as mixtures of them, hopefully suppressing the multiple phase's transitions during intercalation/deintercalation [19,20]. The introduction of a small amount of electrochemically inert elements substituting into the transition metal layers could improve structural stability, producing the variation of lattice parameters without sacrificing energy density and safety [11,13]. Among the possible inactive elements, titanium possesses several advantages due to similar radius and large redox potential difference with respect to Mn and the remarkable gap in Fermi level between Ti$^{4+}$ and Ni$^{2+}$/Mn$^{4+}$ [11,17]. For similar layered materials, Cu and Mg ions improved the moisture stability and the cyclability of P2-type layered oxides, respectively [13]. In addition, Mg$^{2+}$ as a dopant enabled the cathode to attain high coulombic efficiencies and stable long-term cycling. On the other hand, an MgO coating can act as a hydrogen fluoride (HF) scavenger and can protect the cathode surface when exposed to the electrolyte [15]. Alongside the proper synthesis method, the search for new kind of dopants, as well as the right amount of them, is an important parameter that can be useful to improve the electrochemical properties of electrode materials.

In the present paper, pure and doped (Cu, Ti, and V) NaNi$_{0.5}$Mn$_{0.5}$O$_2$ samples were synthesized by means of sol–gel synthesis to determine the influence of the chosen dopants on the phase stabilization. Wide use was made of X-ray powder diffraction with Rietveld structural refinements, also applied on post-mortem electrodes. A look was addressed also to the air stability, a well-known issue for layered materials. Cyclic voltammetry and charge–discharge measurements were performed to evaluate the effect of doping on capacity values, capacity retention, and long-term cyclability.

## 2. Materials and Methods

### 2.1. Synthesis

Pure NaMn$_{0.5}$Ni$_{0.5}$O$_2$ and doped NaMn$_{0.45}$Ni$_{0.45}$M$_{0.1}$O$_2$ (with M = Ti, Cu, V) were synthesized by sol–gel method. NaNO$_3$ (10 wt% excess), Mn(NO$_3$)$_2$ 4H$_2$O, Ni(NO$_3$)$_2$ 6H$_2$O, and Cu(NO$_3$)$_2$ 6H$_2$O, vanadium acetyl-acetonate (C$_{15}$H$_{21}$O$_6$V), or titanium oxide bis(2,4 pentanedionate) (C$_{10}$H$_{14}$O$_5$Ti) were weighted in the proper amount and dissolved in the minimum water amount together with citric acid (2:1 moles with respect to the sum of those of cations). The solution was stirred, then heated at 70 °C and allowed to dry overnight. The obtained powder was treated in oven for 3 h at 300 °C to remove the organic components, then, the powder was pelletized and heated at 900 °C for 12 h (heating rate 5 °C/min, cooling rate 25 °C/min). Cu-doped sample was instead treated at 800 °C for 12 h. All the obtained samples were preserved in a glovebox (MBraun, O$_2$ < 1 ppm, H$_2$O < 1 ppm) to avoid the contact with atmosphere until the subsequent use. Henceforth, NaMn$_{0.5}$Ni$_{0.5}$O$_2$, NaTi$_{0.1}$Mn$_{0.45}$Ni$_{0.45}$O$_2$, NaV$_{0.1}$Mn$_{0.45}$Ni$_{0.45}$O$_2$, and NaCu$_{0.1}$Mn$_{0.45}$Ni$_{0.45}$O$_2$ will be named "NMNO," "NMNO-Ti," "NMNO-V," and "NMNO-Cu" respectively.

Another sample with the NaTi$_{0.1}$Mn$_{0.4}$Ni$_{0.5}$O$_2$ stoichiometry was prepared with the same method and treated to the same temperatures of the analogue NMNO-Ti. In the following sections, this sample is named "NMNO-Ti-def."

## 2.2. Instruments

X-ray powder diffraction (XRPD) measurements were performed using a Bruker D5005 diffractometer (Karlsruhe, Germany) with CuK $\alpha$ radiation (40 kV, 40 mA), graphite monochromator and scintillation detector. The patterns were collected in the angular range of 11–50°, with a step size of 0.03° and counting time of 3 s per step in a silicon sample holder with low background. Rietveld structural and profile refinement was carried out by means of TOPAS (Version 3.0, Bruker, Karlsruhe, Germany) software [21] on the basis of the known P and O crystal structure models for layered materials [1,4]. During the refinement, the background coefficients, scale factor, zero or displacement error, lattice parameters, crystallite sizes, isotropic thermal factors, and atomic positions were varied. The occupancies were fixed to stoichiometric values, due to the small differences between the scattering power of transition metals. To verify the moisture stability of the materials, XRPD patterns were also collected after storage in air.

A Zeiss EVO Ma10 (Carl Zeiss, Oberkochen, Germany) scanning electron microscope (SEM) was used for the morphological study on gold-sputtered powder samples and on the slurries for electrochemical measurements. EDS microanalysis was also performed using the X-ray detector (X-max 50 mm, Oxford Instruments, Wiesbaden, Germany) on the as prepared sample powders.

The slurries for the electrochemical measurements were prepared by mixing the active materials with carbon (Super C65) and binder (PVdF) in a weight ratio of 80:10:10 in N-methyl-2-pyrrolidone (NMP, Aldrich) and magnetically stirred for half an hour. Afterwards, the slurries were coated on an aluminum foil using a homemade doctor blade, maintained 2 h in a vacuum oven at room temperature to remove a large part of solvent, and then dried in the same oven at 100 °C for 1 h and hot pressed with 200 psi at 95 °C for 5 min. For Cu-doped sample, many difficulties were encountered to obtain a homogeneous liquid dispersion with this procedure, so PVdF and carbon were mixed in a mortar, added to NMP, and stirred, and after some time, the active material was added to it. The homogeneity of this slurry was verified with SEM image (Figure S1).

The electrodes were cut in form of discs (1 cm of diameter) with mass loading of about 4 mg/cm². Swagelok cells were assembled in a dry box under a argon atmosphere (MBraun, $O_2$ < 1 ppm, $H_2O$ < 1 ppm) with the slurries as working electrode, Na metal as reference and counter electrode, and a Whatman GF/A disc as the separator. The electrolyte of choice was 1 M $NaClO_4$ in propylene carbonate (PC).

Cyclic voltammetry (CV) was performed using an Autolab PGSTAT30 (Eco Chemie, Metrohm, Utrecht, The Netherlands) at a scan rate of 0.1 or 0.05 mV/s for six cycles in the potential range of 2.0–4.5 V. For galvanostatic charge–discharge tests, the Swagelok cells were cycled on a Neware (Hong Kong, China) battery tester in the same potential range for 10 cycles at C-rates between C/10 and 2C, after a conditioning cycle at C/20. The most promising sample was also cycled for 200 cycles at C/5 (with 5 cycles at C/10 before) in the same potential window.

## 3. Results and Discussion

### 3.1. X-ray Powder Diffraction

In Figure 1, the patterns of pure and doped samples are reported; the differences between them suggest the stabilization of different phases/polymorphs. In general, the samples have a low degree of crystallinity, apart from NMNO-Ti. The most characteristic peak of the layered phases, the 003 reflection at about 16.5°, is clearly shifted or even split into at least two components in the different samples, suggesting that more than one layered oxide coexist [1,6]. The patterns of the two Ti-doped samples are stackable (apart from a peak at about 44° due to NiO) and some similarities can be found also with NMNO-V, having, however, much lower intensities of the reflections. Some common points can also be found between NMNO and NMNO-Cu. The pure NMNO sample is constituted by a mixture of polymorphs, differently from what commonly is reported in the literature, when it is often stabilized as a single O3 phase [8,9]. On the other hand, it is also reported

that the phase stabilization is strictly dependent on the experimental parameters such as the temperature of thermal treatment and the use of quenching or heating/cooling rates [1]. Therefore, this could be the reason of the structural difference of our sample with respect to the literature ones, as well as the exact sodium stoichiometry. In Figure S2, the patterns collected on NMNO samples obtained at different experimental conditions are reported. No significant differences are observed between them, apart from some small additional peaks or slightly different peak intensity ratio, particularly evident in solid-state sample.

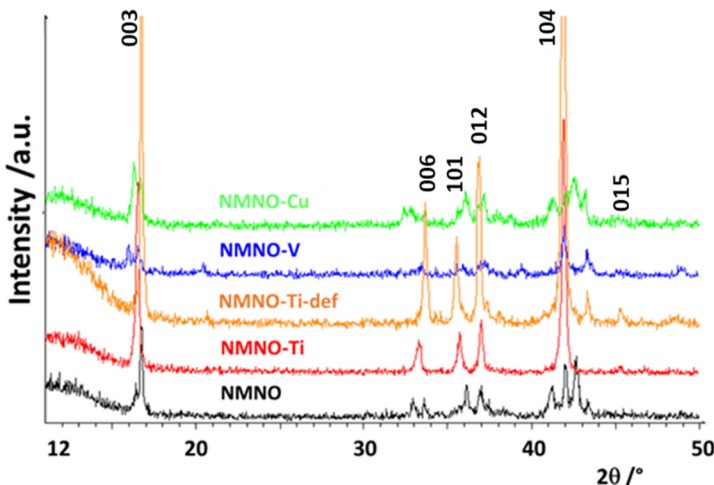

**Figure 1.** XRPD patterns of pure and doped samples. The Miller indices of the main reflections of the O3 polymorph are also reported.

The complex phases' equilibrium was expected and desired; in the literature, the doping is, in fact, suggested as a way to stabilize a peculiar polymorph or a mixture of them, with the aim to reduce the structural transitions that cause the electrochemical performance decay [13,20,22].

In order to precisely determine and quantify the phases in the different samples, structural refinements based on the Rietveld method were performed. The structural models of the different polymorphs were taken from the literature; the calculated patterns expected for the main O and P type phases are reported in Figure S3 [1,2]. As an example, in Figure 2, the refinement results on NMNO-Ti pattern are shown. The experimental curve (blue) is compared with the calculated one (red), and in the bottom, the difference curve (grey) and the bars of the expected peak positions of the different phases are also shown. Good fitting results are obtained; the only imperfection concerns the peak at about 16.5°, the 003 reflection, affected by preferred orientation effect, which is difficult to satisfactorily model during the refinements notwithstanding the use of the proper coefficient of orientation [11,20]. The main structural parameters such as zero (or displacement) error, lattice parameters, crystallite sizes, and weight percentages were determined for all the synthesized samples. In Figure 3, the weight percentages and the crystallite sizes values of the different polymorphs/impurities are reported.

For what concerns the weight percentages (Figure 3), it is obvious that NMNO-Ti sample is O3 type monophasic, as also reported in the literature [11,13], as well as NMNO-Ti-def. NMNO is instead constituted by a mixture of polymorphs in similar amount, while vanadium-doped sample is a biphasic mixture of O phases [5]. For Cu doping, mainly O1 and P'3 phases are detected. In some cases, traces of NiO were detected, as also common in the literature [11,13].

The solubility of the different elements into the layered crystal structure is high, demonstrating the versatility of the lattice framework. All the chosen ions (Cu, Ti, and V) were easily accommodated into Mn/Ni octahedral crystallographic sites and no segregation of the corresponding oxides was verified (even in the literature NiO is considered a ubiquitous secondary phase). The main issues were detected with Cu-doped samples,

which required a peculiar thermal treatment consisting of multiple steps at increasing temperature, probably because of the higher stability of Cu ions in a different coordination (i.e., planar).

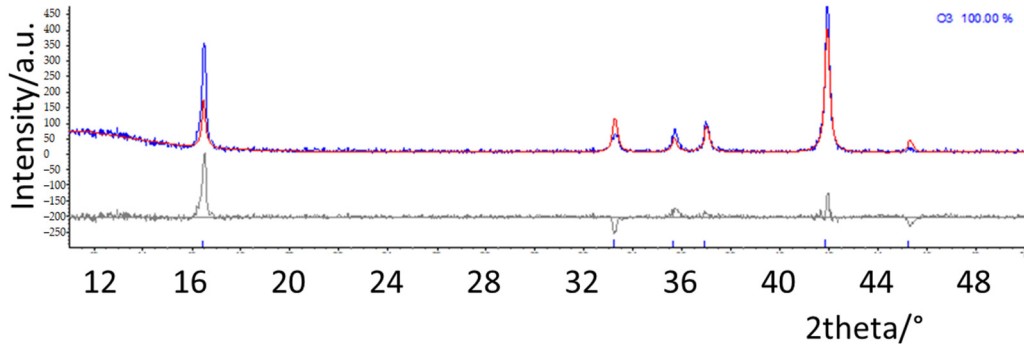

**Figure 2.** Rietveld refinement of NMNO-Ti pattern. The experimental curve (blue) is compared with the calculated one (red), and in the bottom, the difference curve (grey) and the bars of the expected peak positions of the phases are also shown.

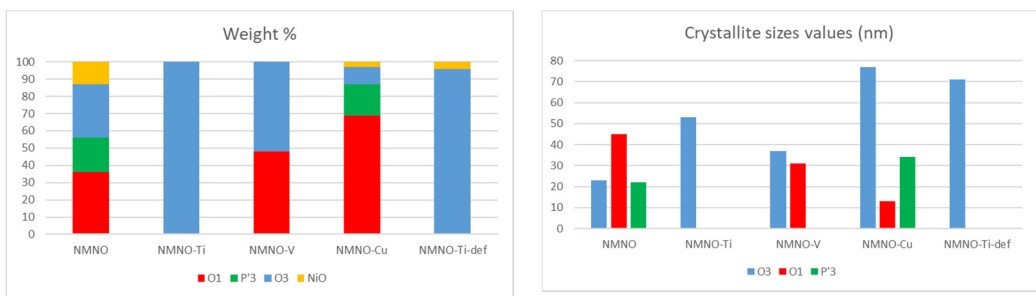

**Figure 3.** Weight percentages and crystallite sizes values obtained from Rietveld refinements of the different polymorphs for all the samples.

The crystallite sizes values (Figure 3) range between 35 and 75 nm. In a single sample, the values for the coexisting polymorphs can vary and the crystallite sizes values of a single polymorph change depending on the involved sample. However, from these results, it seems that the crystallite sizes do not have a peculiar trend with doping.

The layered oxides are known to be highly unstable in air, due to the ease of reaction with moisture [13], and therefore, it is mandatory to stock them in dry box. This happened also for our samples: undoped and, unfortunately, all the doped samples converted in some days into "hydrated phases" [9,13]. For these reasons, we avoided drying the slurries in air, however, this step was performed in vacuum oven. The doping, differently from other systems [13], does not help to improve the air stability. In Figure S4, as an example, the XRPD patterns of NMNO-Ti sample stored in a dry box and in air are reported. The decrease in crystallinity and the formation of a peak at about 12°, typical of "hydrated" phases, are evident [9].

### 3.2. Scanning Electron Microscopy

In Figure 4, the micrographs of all the samples are shown.

Extended aggregates, particularly for NMNO and NMNO-Ti-def, of fine particles are evident. This could be in line with the XRPD observations, evidencing crystallite sizes values under 80 nm. NMNO-Ti is constituted by compact blocks, on which the lightest and flake-like particles are deposited, resembling the stacking layers of octahedra forming the O3 structure, the only one present in this sample. NMNO-V sample has instead large particles, again aggregated in wide blocks and also in this case, flake forms could be guessed: this sample, from XRPD, is in fact constituted by O type phases. Cu-doped sample shows very compact aggregates, in which a melted aspect could be recognized and some

light rounded particles emerge from the block [11,19]. SEM measurements demonstrate that no significant influence of dopant ions is evident on the external morphology, but the main effect can be probably ascribed to the prevalent polymorph, driving the particles growing towards flakes or spheres. The stoichiometries of all the samples were verified by means of EDS micro-analysis. Good results were found for all the samples (see Table 1). As an example, in Figure S5, the EDS spectrum of NMNO-Cu sample is reported.

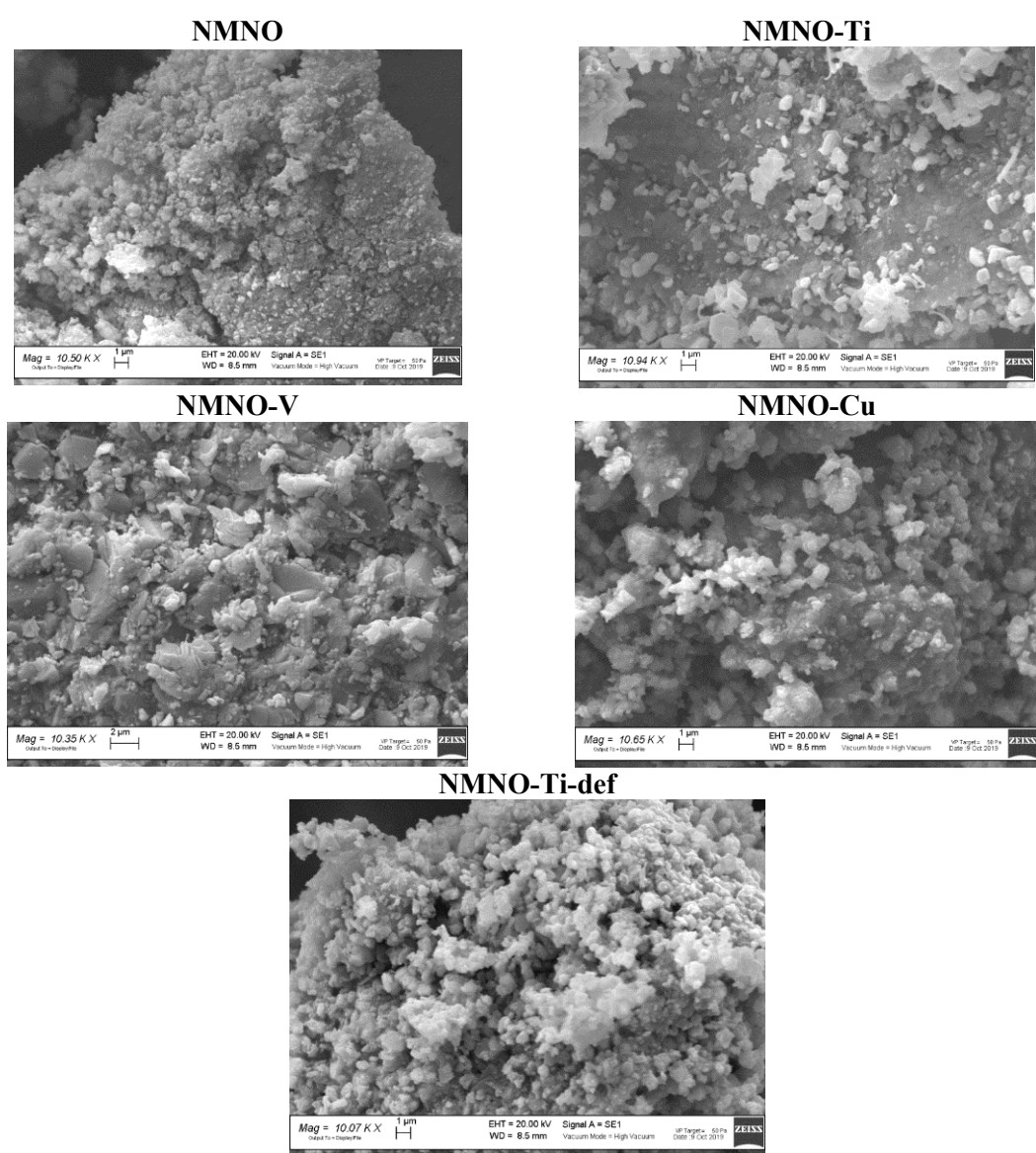

**Figure 4.** SEM micrographs of undoped and doped samples.

**Table 1.** Elemental composition (at %) of all the samples as determined by EDS microanalysis.

| Elements (Atomic %) Sample | Na | Mn | Ni | Dopant |
|---|---|---|---|---|
| NMNO | 23.11 | 11.32 | 10.91 | – |
| NMNO-Ti | 19.32 | 8.57 | 8.50 | 2.09 |
| NMNO-Ti-def | 19.21 | 7.98 | 8.95 | 1.88 |
| NMNO-Cu | 21.03 | 9.34 | 9.42 | 2.21 |
| NMNO-V | 20.54 | 9.12 | 9.22 | 2.15 |

### 3.3. Cyclic Voltammetry (CV)

The pure and doped samples were subjected to six cycles of cyclic voltammetry to identify the redox phenomena and to evidence the eventual effect of dopants.

Multiple peaks fall in the investigated potential range, usually ascribed to $Ni^{2+/3+}$ (2.4–3.3 V) and $Ni^{3+/4+}$ (3.3–4.4 V). Even if these two sets of peaks would match well with the increase in the electrochemical potential of cathodes with the number of electrons in d orbitals of transition metal elements in the same period (meaning that the peaks below 3.3 V can be ascribed to $Mn^{3+/4+}$ and those above 3.3 V to $Ni^{3+/4+}$), different authors agree on the two-electron nature of the redox reaction involving only nickel, as proven by XPS measurements [13,16].

In Figure 5, the six CV cycles, A–E, for all the samples are reported.

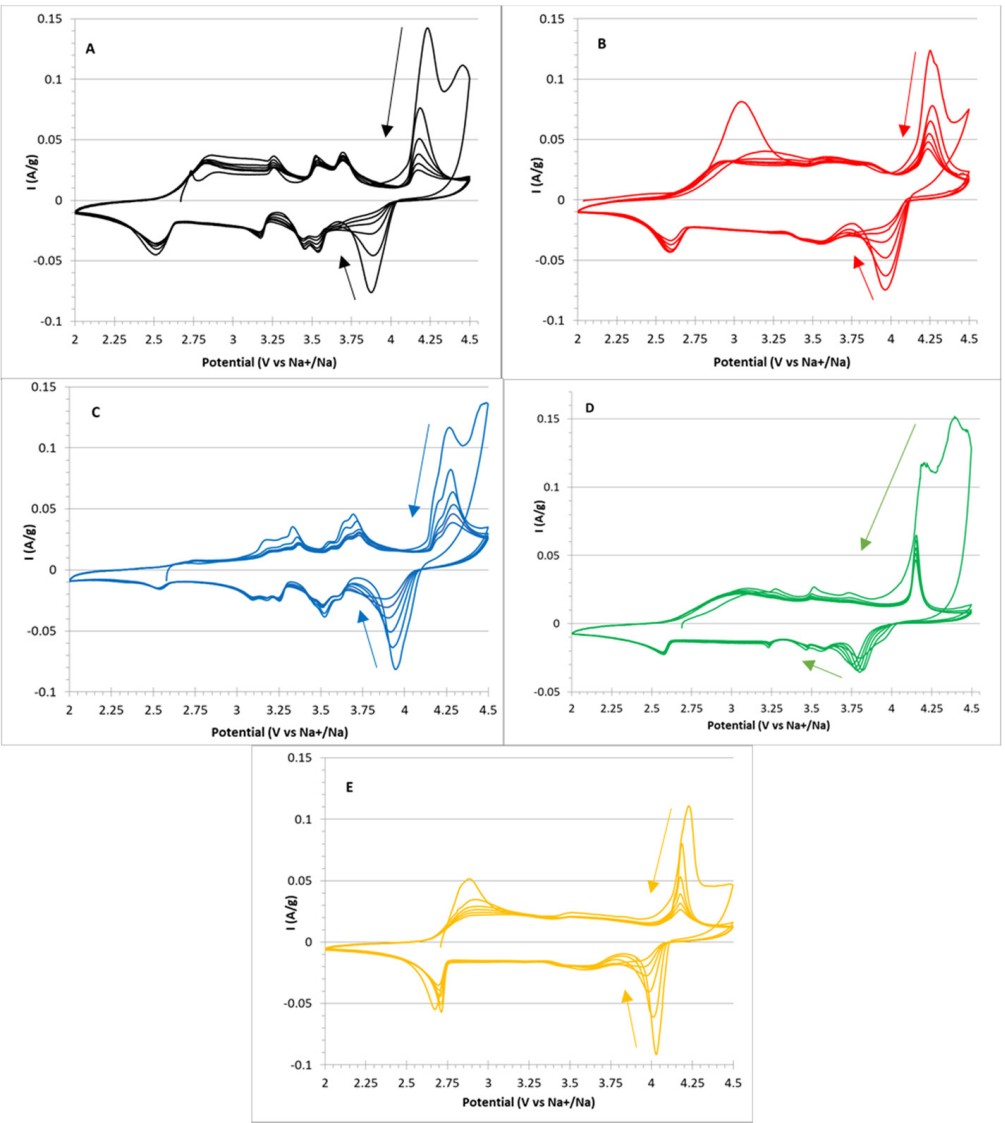

**Figure 5.** Cyclic voltammetry cycles in the 2–4.5 V potential window for (**A**) NMNO, (**B**) NMNO–Ti, (**C**) NMNO–V, (**D**) NMNO–Cu, and (**E**) NMNO–Ti-def samples. The arrows indicate the progression of cycles.

For NMNO (Figure 5A), five peaks can be detected (anodic/cathodic potential): 4.16/3.92, 3.69/3.54, 3.52/3.45, 3.25/3.18, and 2.82/2.52 V, the last one being particularly broad. The peak intensity decreases upon cycling, showing low capacity retention. This is especially true for the highest peak, suggesting problematics related to the full

recovery of Na stoichiometry, which can be probably related to irreversible/little reversible phase transitions. Similar results are reported in the literature, [12,23] even if the cutoff potential is usually set to 4 V.

Doping introduces little changes in the relative intensity ratios, maintaining the overall shape of the voltammogram. It is remarkable that both samples containing titanium cations (Figure 5B,E) show very low first-cycle coulombic efficiency related to the lowest peak, again suggesting an irreversible phase transition occurring in both samples. This is confirmed by the observation that both samples are the only constituted by O3 phase, and among the other samples containing considerable amount of O3 phase, NMNO-V (Figure 5C) is the only sample that has considerable low coulombic efficiency associated with the same peak.

The voltammogram of NMNO-V is very interesting, i.e., all the peaks are highly resolved and split into two components. This could suggest that the vanadium cations contribute to the electrochemical activity, causing a split in the peak potentials (i.e., perturbative effect of the redox couple with respect to the free energy related to the phase transition).

The voltammogram of NMNO-Cu (Figure 5D) shares a peculiar feature with the former (NMNO-V) that the high end of the anodic scan shows very high currents in the first cycle, even higher than the redox peak at 4.25 V. By comparison with the electrochemical stability window of the electrolyte in the other samples, it can be inferred that a further irreversible reaction took place in these two samples. We have discussed about this hypothesis later in the paper.

The third cycles of CV measurements of all the samples are plotted in Figure 6.

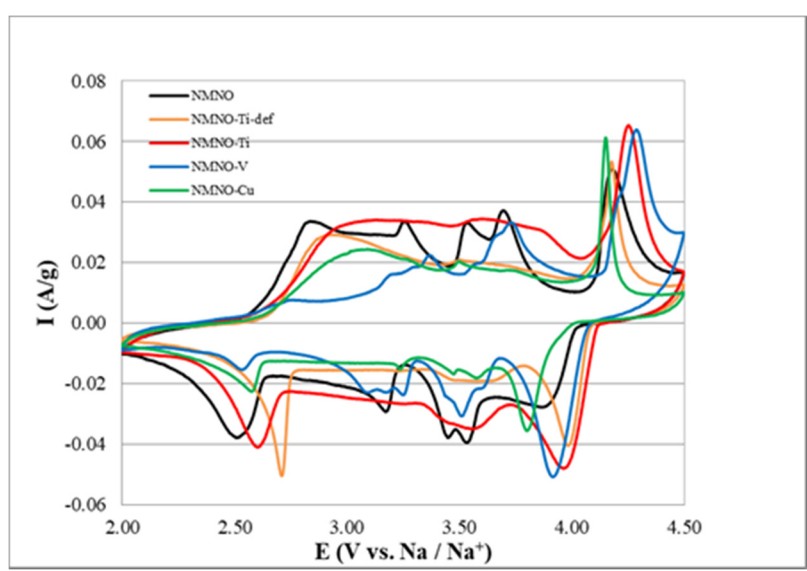

**Figure 6.** Comparison between the third cycle of the CV of all the samples. The curves are normalized for the mass of active material for better comparison.

Broad curves are detected for most samples, except for the aforementioned NMNO-V. In order to obtain better resolution, slower potential scan on the NMNO-Ti def sample (Figure S6) was collected, but the results were not decisive. Peak broadening can be related to the kinetics of electrode reactions, i.e., broader and higher peaks suggest pseudocapacitive contributions to the total current [24–26]. The sharpest redox peaks are shown by NMNO and NMNO-V, which are the ones with the worst cyclabilities, as presented later in the paper. Phase transitions, often associated with sharp redox peaks (or voltage plateaus), have a negative impact on the electrode performance because of the associated reversibility issues and volume changes that can even lead to disgregation of the electrode in other electro-chemical systems.

The behavior of NMNO-Ti def suggests a capacitive contribution, because of the huge intensity decrease in half of the sweep rate (Figure S6), but it still shows broad peaks. This feature reflects better for Ti-doped samples in the cycling performances.

Of course, cation substitutions have an impact on the free energy associated with intercalation/deintercalation reactions, i.e., the peak potentials shift with respect to NMNO with, on average, higher redox potentials. This feature is particularly beneficial for a cathode material to be employed in sodium-ion batteries, where Na metal cannot be used as an anode material due to the formation of dendrites [1–3]. Further, some voltammograms seem to confirm the suppression of polymorphic transitions, with flatter curves in the 2.7–3.2 V region.

### 3.4. Charge–Discharge Measurements

The rate capability test for all the synthesized samples is reported in Figure 7. The best performances, at almost every C-rates, are found for NMNO-Ti [13,23].

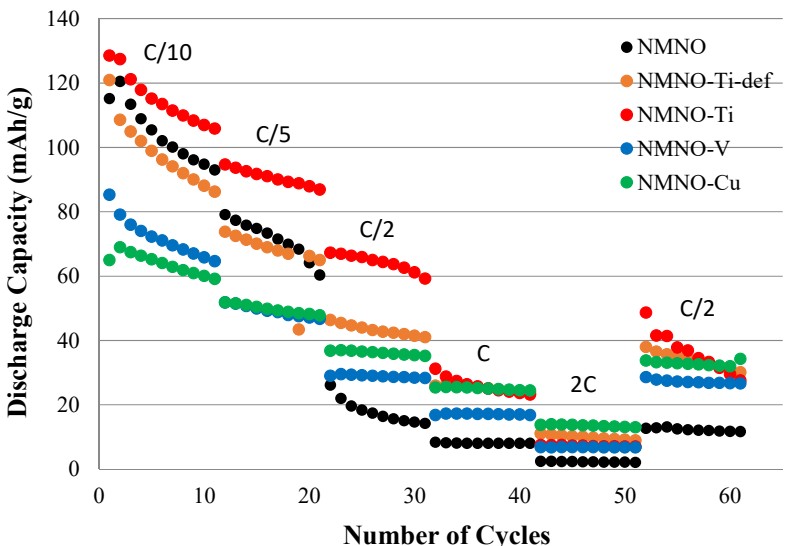

**Figure 7.** Discharge capacity values of all the samples at different C-rate values.

NMNO has comparable performances at C/10 and C/5, which drop at the higher C-rates, probably due to the too slow semi-infinite linear diffusion kinetics associated with phase transitions, also explaining the unsatisfactory results of NMNO-V.

While at higher C-rates, NMNO-Ti, NMNO-Cu, and NMNO-Ti-def have a similar trend with little capacity differences, and the latter show worse performances at slower C-rates [23]. The effect of the cation to be substituted (equi-atomic or only Mn) has no influence on the capacitive contribution prevailing at higher C-rates, but it affects the performances on the solid-state diffusion. The capacitive contribution inferred from CV measurements also explains the similarities between the capacities obtained between the aforementioned three samples.

The capacity recovery at C/2 after faster cycling is almost perfect for all the samples except for NMNO-Ti, which shows lower values. This can suggest irreversible phase transitions negatively affecting the electrochemical behavior. It is well-known, in fact, that O3 phase converts to P3 during cycling, showing low capacity values, as determined in our case.

The sample with the best performance, NMNO-Ti, has been subjected to a prolonged cycling (200 cycles at C/5) (Figure 8). The Coulombic efficiency is very good (>97%), confirming the positive effect of titanium onto electrochemical performance.

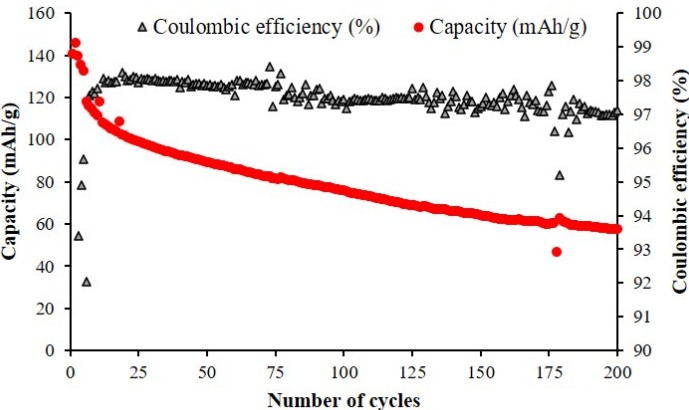

**Figure 8.** Discharge capacity values for 200 cycles at C/5 (after 5 cycles at C/10) for NMNO-Ti sample.

*3.5. Ex-Situ XRPD Analysis*

To verify the inferred structural transitions during charge–discharge measurements, ex-situ XRPD patterns were collected after cycling (Figure 9). The Swagelok cells were opened, and the cathodes were rinsed with ethanol to remove sodium salt traces and separator fibers. In all the patterns, the peak of Al (the current collector) was marked. Many differences with respect to the initial mixture of phases can be appreciated. First, the patterns are suggestive of crystalline samples: the patterns of NMNO and NMNO-Ti are similar, with only some peak shifts and larger peaks in the first, while NMNO-Cu and NMNO-V present some differences. All the phases were recognized and quantified by applying the Rietveld refinement (Figure S7). We should take into consideration that the absolute amount of phases are biased by the presence of the aluminum phase (the current collector), but the relative amount of phases are still valid.

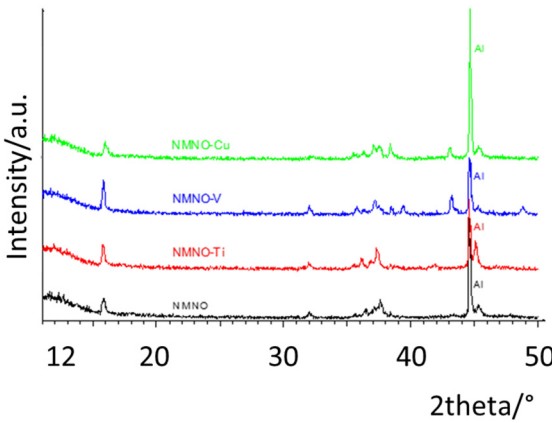

**Figure 9.** Ex-situ XRPD patterns collected after charge–discharge measurements for all the samples.

The main phase, for all the samples, is P3, with traces of remnant O3 [1,23]. The O3 phase is characterized by the presence of a peak at about 17°, as well as the absence of the intense peak at about 42°. For NMNO, originally formed by a mixture of P and O phases, the presence of P1 phase is also detected, as well as for NMNO-Cu, where also NiO is found. NMNO-V sample is the only one showing P'2 as the main phase, together with P3 and NiO.

**4. Conclusions**

The effectiveness of doping in the improvement of electrochemical performance of NaNi$_{0.5}$Mn$_{0.5}$O$_2$ was well demonstrated in the present paper. The dopant ions substitute Ni/Mn on their octahedral crystallographic sites, producing different stabilization of poly-

morphs, with the O3 phase as the most represented. This has an impact on voltammograms, which show broad peaks, associated with multiple structural transitions. The contribution of the dopant ions to the redox reactions, particularly for vanadium doping, cannot be excluded. The best performances are shown by Ti-doped sample with the highest capacity values (120 mAhg$^{-1}$ at C/10); however, at higher currents (1C), Cu-substituted sample also has stable and comparable capacity values.

The right arrangement of dopant amount and type, the kind of synthesis, and temperature and time of treatment could be the key for obtaining optimized samples with good and stable capacity values for their application in sodium batteries.

**Supplementary Materials:** The following are available online at https://www.mdpi.com/article/10.3390/electrochem2020024/s1, Figure S1: SEM image of the slurry of NMNO-Cu sample; Figure S2: Comparison between NMNO patterns obtained in different experimental conditions; Figure S3: Calculated models of the main O and P layered polymorphs, used to perform the Rietveld structural refinements described in the paper; Figure S4: XPRD patterns of NMNO-Ti before (black) and after (blue) air maintenance for some days; Figure S5: EDS spectrum; Figure S6: (A) Three cycles of CV at 0.05 mVs$^{-1}$ and (B) comparison between the third cycle of the cyclic voltammetry at 0.01 mVs$^{-1}$ (continuous line) and 0.05 mVs$^{-1}$ (dotted line) for NMNO-Ti sample; Figure S7: Rietveld refinements of ex-situ patterns collected post-mortem on the electrodes.

**Author Contributions:** Conceptualization, M.B. and D.S.; methodology, F.L. and D.N.; formal analysis, F.L. and D.N.; investigation, M.A.; writing—original draft preparation, M.B. and D.S.; writing—review and editing, M.B., D.S., and M.A.; supervision, M.B. All authors have read and agreed to the published version of the manuscript.

**Funding:** This research received no external funding.

**Institutional Review Board Statement:** Not applicable.

**Informed Consent Statement:** Not applicable.

**Data Availability Statement:** The data presented in this study are available on request from the corresponding author. The data are not publicly available due to privacy.

**Acknowledgments:** We thank Alessandro Girella for the collection of SEM images and Irene Quinzeni for the fruitful discussion.

**Conflicts of Interest:** The authors declare no conflict of interest.

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
