# Peer review of "Synergistic Effect of Polymorphs in Doped NaNi0.5Mn0.5O2 Cathode Material for Improving Electrochemical Performances in Na-Batteries"

_2673-3293, doi:10.3390/electrochem2020024_

Round 1
Reviewer 1 Report
Reviewer’s comments
Manuscript Number: electrochem-1217289
Title: Synergistic effect of polymorphs in doped NaNi0.5Mn0.5O2 cathode material for improving electrochemical performances in Na-batteries
Journal: Electrochem
The manuscript reports the synthesis of NaNi0.5Mn0.5O2 cathode material for Na-batteries. The manuscript has enough data and well written. However, some comments need to be addressed as follows:
- All XRD peaks need to be identified, and their corresponding plans need to be highlighted.
- Since the material contains many elements in different ratios (Na, Ni, Mn, O, Ti, Cu, V), elemental (EDX) analysis is needed to confirm the component ratios.
- The crystallite size obtained from XRD should be compared/correlate to the size from SEM analysis.
- More attention is needed to the formatting, such as chemical formulas.
- The main findings (values) should be highlighted in the conclusion.
- The discussion on the energy storage associated with intercalation/deintercalation needs deep explanation, the literature on the energy storage applications of mixed metal oxide may help.
- The supplementary file still contains the author's comments ( S5); this should be removed.
Based on the comments given above, the current manuscript needs a major revision.
Reviewer 2 Report
1) The paragraph on page 2 line 67 is improper. Operando techniques are not used in this study.
2) There is lack of discussion on why NaTi0.1Mn0.45Ni0.45O2 outperform NaTi0.1Mn0.4Ni0.5O2 with the same amount of Ti doping.
3) Although showing improvement in electrochemical performance, the distinct effect of Ti, V, and Cu doping on NaNi0.5Mn0.5O2 is unclear. What are the doping sites in the crystal structure? Why are different phases formed?
4) The specific capacity of NMNO-Ti keeps decrease upon cycling in Figure 8. The capacity retention is not good. What about the cycling performance of other samples? The effects and mechanisms of Ti doping, which is claimed to be more favourable than others, need to be confirmed and analysed.
Reviewer 3 Report
This paper systematically investigated the effect of different dopants on the structure and electrochemical performance of NaNiMnO. It could be accepted after major revision.
(1) Some typos should be revised. Such a LIne 96, should be "Cu Kα". The references are double numbered.
(2) The biggest problem of this paper is that it is not written in scientific language, which makes it rather confusing and difficult to understand. Do not just describe controversial conclusions from references, please clearly state results from your experiment. One example is the description at Line 140-144, no idea what's the result of this paper. If the experiment condition can affect the structure of the product, the Cu doped sample is not comparable with other samples since it uses different heating temperature.
(3) Please explain the purpose of Ti-def sample.
(4) Since at line 63, the author stated Cu doping can improve the stability of oxides, but at line 198-199. Cu doping obviously does not improve stability. PLease explain why.
(5) The Introduction part should be re-organized to emphasize the importance and significance of this work.
(6) Still confusing the effect of dopants on the structures and electrochemical performance. Better to re-organize the conclusion part.
(7) Figure 8 the color of two symbols are too close, difficult to distinguish.
Reviewer 4 Report
Article ID: Electrochem 1217289
Article type: Research Paper
Title: Synergistic effect of polymorphs in doped NaNi0.5Mn0.5O2 cathode material for improving electrochemical performances in Na-batteries
Authors:Dr. Leccardi et al.
In their manuscript, Dr Leccardi et al. describe the influence of doping in sodium layered oxide materials. They studied the influence of different metal ion in the Na-batteries performance. The morphology and electrochemical properties were investigated to determine the most relevant parameters using several techniques. The study is quite interesting and the manuscript is well-written. Thus, I recommended this manuscript for publication in Electrochem once the authors revise it by addressing the following minor points:
- Some grammatical errors are present and the manuscript should be corrected. ex: line 137 “over imposable” replace by “stackable”, line 247 “contribution” replaces by “contribute”, etc.
- In the figure 5, it is not obvious to see the evolution of electrochemical waves after cycling. Some arrows should be inserted to show the evolution. Are all peaks decreased ?
- In line 258 the paragraph is not clear. what did the authors mean with peak broadening ? which peaks ? even the NMNO showed such behaviour. The conclusion of this part is not clear. May be the authors can discussed a little bit more.
- the numbering of SI figures did not correspond to the manuscript.
- one suggestion: if the peak broadening is related to the kinetics of the reaction. the plot of peak current vs scan rate can give us a good information. The authors started to study the scan rate influence but did not plot this evolution.
Round 2
Reviewer 1 Report
The authors have addressed my comments in the revised manuscript (electrochem-1217289).
Reviewer 2 Report
The authors have revised the manuscript according my comments.
Reviewer 3 Report
Minor typos should be corrected.